# Adolescent Smartphone Overdependence in South Korea: A Place-Stratified Evaluation of Conceptually Informed AI/ML Modeling

**DOI:** 10.3390/ijerph22101515

**Published:** 2025-10-02

**Authors:** Andrew H. Kim, Uibin Lee, Yohan Cho, Sangmi Kim, Vatsal Shah

**Affiliations:** 1School of Social Work, Rutgers, The State University of New Jersey, New Brunswick, NJ 08901, USA; ahjkim92@rutgers.edu (A.H.K.); vatsal.b.shah99@rutgers.edu (V.S.); 2Department of Human Development and Family Studies, The University of Alabama, Tuscaloosa, AL 36849, USA; 3Department of Community, Family, and Addiction Services, Texas Tech University, Lubbock, TX 79415, USA; yohan.cho@ttu.edu; 4College of Social Work, University of Tennessee, Knoxville, TN 37996, USA; skim153@vols.utk.edu

**Keywords:** smartphone overdependence, adolescents, conceptually informed modeling, artificial intelligence (AI), machine learning (ML), low-risk screening tools, construct-level analysis, eXplainable AI (XAI), urbanicity/place-based analysis, South Korea

## Abstract

Smartphone overdependence among South Korean adolescents, affecting nearly 40%, poses a growing public health concern, with usage patterns varying by regional context. Leveraging conceptually informed AI/ML models, this study (1) develops a high-performing low-risk screening tool to monitor disease burden, (2) leverages AI/ML to explore psychologically meaningful constructs, and (3) provides place-based policy implication profiles to inform public health policy. This study uses data from 1873 adolescents in the 2023 Smartphone Overdependence Survey by the National Information Society Agency (NISA) in South Korea. Across the sample, the adolescents were about 14 years old (SD = 2.4) and equally distributed by sex (48.1% male). We then conceptually selected 131 features across two domains and 10 identified constructs. A nested modeling approach identified a low-risk screening tool using 59 features that achieved strong predictive accuracy (AUC = 81.5%), with Smartphone Use Case features contributing approximately 20% to performance. Construct-specific models confirmed the importance of Smartphone Use Cases, Perceived Digital Competence and Risk, and Consequences and Dependence (AUC range: 80.6–89.1%) and uncovered cognitive patterns warranting further study. Place-stratified analysis revealed substantial regional variation in model performance (AUC range: 71.4–91.1%) and distinct local feature importance. Overall, this study demonstrated the value of integrating conceptual frameworks with AI/ML to detect adolescent smartphone overdependence, offering novel approaches to monitoring disease burden, advancing construct-level insights, and providing targeted place-based public health policy recommendations within the South Korean context.

## 1. Introduction

Smartphones have become deeply embedded in modern life, particularly for adolescents, who rely on them for education, entertainment, and social interaction. In South Korea, nearly all adolescents (99.1%) owned a smartphone in 2024 [1], and screen time has continued to rise at an estimated annual growth rate of 12.6% since 2016 [2]. By 2021, adolescents reported approximately five hours of daily smartphone use [3]. This mass exposure and rising engagement have fueled growing concerns about excessive and poorly controlled use. Researchers have conceptualized this phenomenon as a potential behavioral addiction [4,5], emphasizing impaired self-regulation, functional impairment, and parallels with other behavioral addictions. This aligns with the International Classification of Diseases (ICD)-11, which highlights loss of control and persistence despite adverse consequences as defining features of addictive disorders [6]. In South Korea, government policy formally defines this as smartphone overdependence, a state in which excessive use reduces regulation and produces problematic consequences [7]. Smartphone overdependence is defined as a maladaptive state characterized by salience, impaired self-regulation, and negative social, physical, and academic consequences, and it is measured through the validated Smartphone Overdependence Scale used in Korea’s national surveys [8]. This construct is theoretically and clinically relevant as impulsivity and emotional dysregulation have been identified as core mechanisms [4,9,10], while personality characteristics and mental health vulnerabilities further increase risk [11]. Empirical evidence also demonstrates associations with poor sleep recovery and physical inactivity [12], disordered eating behaviors [13], and heightened suicide risk [14]. These findings clarify that smartphone overdependence extends beyond screen time, representing a significant behavioral health concern that warrants policy and clinical attention. The Smartphone Overdependence Scale, widely used in public health monitoring, captures tolerance, withdrawal, and daily life disturbance, reflecting a marked decline in self-control such that smartphone use becomes the most salient activity in daily life [15].

The NISA report [7] estimates that 37% to 40% of Korean adolescents between the ages of 10 and 19 meet the criteria for smartphone overdependence, the highest among all the age groups. These prevalence estimates are particularly noteworthy given consistent associations with depression, anxiety, obsessive–compulsive disorder (OCD), attention deficit hyperactivity disorder (ADHD), stress, loneliness, suicidal ideation, and substance use [16,17,18]. A meta-analysis also found negative effects on academic achievement [19], while more recent studies report poor sleep and reduced recovery from fatigue [12], heightened risk of depression and suicidality [14], and co-occurring maladaptive behaviors such as food addiction and emotional eating [13]. At the personality level, adolescents at risk often exhibit higher neuroticism and lower self-control [11]. Collectively, these studies converge on the conclusion that adolescent smartphone overdependence produces wide-ranging harms across the clinical, educational, and public health domains.

In response, South Korean policy emphasizes early detection, monitoring, and intervention, with schools playing a central role in screening [20]. These policies parallel global trends across several countries, where, in France, smartphones are prohibited in classrooms for students under 15 [21], while at least 18 U.S. states restrict smartphone use during the school day [22]. In Korea, schools administer structured self-report screening tools to classify students by risk level and connect them with interventions such as psychological counseling, parental education, and behavioral therapy, reflecting a coordinated multi-system response [23].

Despite these efforts, the current approaches rely heavily on lengthy self-report surveys that require multiple reporters and include psychologically intrusive questions. For example, items often ask whether students feel hopeless without their phones, experience declines in academic performance, or feel depressed after extended use [24]. While designed to identify risk, such questions can be distressing for adolescents, who may have less capacity than adults to tolerate self-reflection on sensitive topics. As a result, questions that appear benign to adults may feel intrusive or embarrassing to youth, potentially triggering anxiety or disengagement. Evidence shows that intrusive surveys increase response bias, reduce accuracy, and lead to higher dropout rates [25,26,27]. Ethical frameworks reinforce these concerns, where the Belmont Report emphasizes beneficence and minimizing harm [28], and the APA Code of Ethics requires psychologists to avoid or reduce harm to participants [29]. Empirical work supports these cautions: Hasking et al. [30] found that 15% of adolescents with poorer psychological functioning reported distress during survey completion, with declines in well-being afterward. These findings point to a methodological and ethical gap: the very tools used to detect overdependence may inadvertently cause harm.

Given these limitations, predictive modeling using artificial intelligence and machine learning (AI/ML) has gained attention as a promising alternative. AI/ML technologies, already established in diverse domains [31], are increasingly recognized for their potential as efficient and sensitive mental health screening tools [32]. Unlike traditional surveys, predictive models can achieve high accuracy with fewer and less intrusive indicators. For example, Kim and Song [33] achieved 87.6% precision in predicting adolescent smartphone overdependence using multi-year survey data, while Lee and Kim [34] obtained high performance with a compact set of loggable smartphone-related variables. Other international studies confirm strong performance using psychological and demographic features [35,36]. However, to date, no study has developed a low-dimensional AI/ML model tailored specifically to adolescents or explicitly incorporated conceptually informed ethically minimal feature sets designed to reduce burden and discomfort.

A second research gap concerns contextual variation. Sapienza et al. [37] found that urban residents spend more total time on smartphones, often for productivity, whereas rural residents spend proportionally more time on entertainment. In Bangladesh and Korea, higher population density has been linked to elevated risk [38], while research in China found rural adolescents more vulnerable, citing loneliness and anxiety [39,40]. These contradictions likely reflect differences in samples, measures, operational definitions, and cultural environments. To capture contextual differences in South Korea, this study adopts the government’s official administrative classifications of urbanicity, distinguishing metropolitan cities, medium/small cities, and towns/rural districts under the Local Autonomy Act [41], thereby ensuring that our analysis is aligned with national survey frameworks and directly relevant to policy contexts. Together, they underscore the need for models that are not only accurate but also contextually sensitive and culturally grounded.

In South Korea, where nearly all adolescents own smartphones and the national policy formally recognizes overdependence, the development of such models is both urgent and essential. The present study directly responds to this need by integrating conceptual frameworks with AI/ML to create adolescent-friendly, ethically minimal, and place-sensitive approaches to detection and policy. The specific aims and contributions are outlined in the following section.

### Research Aims and Contributions

This study aims to advance the understanding, detection, and policy implications of adolescent smartphone overdependence in South Korea through the integration of conceptual frameworks and AI/ML methods. Grounded in public health and the social sciences, we outline three aims:Develop a high-performing low-risk screening tool to detect adolescents at risk of smartphone overdependence using conceptually informed AI/ML modeling;Leverage AI/ML toward construct exploration of contributing factors;Provide place-based policy profiles that offer insights into local variations in risk factors and model performance, enabling targeted intervention.

The study contributes methodologically by introducing a framework that prioritizes conceptually meaningful constructs over purely data-driven feature sets. Practically, it informs the design of adolescent-friendly screeners aligned with ethical research principles. Conceptually, it shows how AI/ML can illuminate patterns in adolescent behavior that align with psychologically meaningful constructs, offering new directions for behavioral research. From a policy perspective, it advances place-sensitive approaches to digital well-being, providing insights that can support regionally adaptive public health interventions.

## 2. Materials and Methods

The following sections outline our data, approach to feature selection and grouping, and analyses.

### 2.1. Data

Secondary data, collected by the National Information Society Agency (NISA) in South Korea, was used. The NISA Smartphone Overdependence Survey is a nationally representative household survey of individuals aged 3–69 years in South Korea based on probability weighting. The full sample consisted of 22,844 household members who had used a smartphone within the past month and were present and available for the interview, which took place from September to November 2023. During the interview, each participant was asked approximately 192 questions covering demographic characteristics, smartphone usage, online video services, counseling and education, awareness of overdependency, and psychosocial characteristics. In the current study, we include all 1873 adolescent participants aged 10–18 from the full sample. The upper threshold of age 18 was selected since it marks the final year of high school in South Korea and avoids conflating adolescent experiences with those of emerging adults. College, typically age 19 in Korea, has been shown to signal a profound shift in lifestyle, social environment, academic context, and resultant smartphone use patterns [42]. This age range is consistent with other Korean epidemiological and behavioral health studies demonstrating that smartphone use and its psychological consequences are most relevant within school-aged populations [16,43,44].

### 2.2. Outcome

The outcome variable in this study is smartphone overdependence. This variable was measured with the 10-item Smartphone Overdependence Scale developed by Korea’s National Information Society Agency [45]. The Smartphone Overdependence Scale assesses three domains: (1) self-control failure, reflecting difficulties in regulating smartphone use; (2) salience, including concentration disruption due to smartphone use; and (3) serious consequences, such as health, academic, or interpersonal conflicts caused by use. The scale comprises a 4-point Likert scale ranging from strongly disagree (1) to strongly agree (4). Based on total scores, adolescents are categorized as high-risk (≥31 points), potential risk (23–30 points), or no risk (<23 points), consistent with prior Korean adolescent work [46,47]. In our AI/ML models, we use the binary any-risk versus no-risk target as our label by combining the potential risk and high-risk groups. This is in line with our research aim to develop a low-risk screener for public health detection. The Smartphone Overdependence Scale has been validated in large nationally representative adolescent samples. In the 2020 Korea Youth Risk Behavior Survey (N = 54,948), Cronbach’s α was 0.92 [47], and the original development study reported α = 0.84 [45], further supporting reliability across populations. Convergent validity has been demonstrated through consistent associations with stress, low sleep quality, unhappiness, sadness/despair, and loneliness—patterns theoretically aligned with overdependence [45]. The Cronbach’s α in the present study sample was 0.86.

### 2.3. Conceptual Feature Selection and Construct Grouping

From the NISA survey, all individual items related to adolescents were considered candidates for inclusion. While the original NISA survey includes five broad domains (smartphone use, online video use, prevention and counseling, problem perception, and psychosocial characteristics), we regrouped items based on theoretical grounding. Two overarching domains (demographics and urbanicity) and ten theoretically grounded constructs were identified. Only items aligning with one of these domains or constructs were retained for analysis. After one-hot encoding, we removed fourteen features with missingness due to skip logic and one feature due to never being endorsed in the sample. In the case of skip logic missingness, other retained features were determined to provide conceptual overlap. For example, the feature “do you think the content you typically use on smartphone video streaming services is primarily entertainment?” would overlap with the retained binary indicator of “TV/Entertainment as the #1 mostly used video content”. Other examples include Likert scale questions about their experience with smartphone preventive education. In these cases, a “0” fill approach would not be appropriate and would substantially restrict the sample (69% of adolescents never attended a preventive program). Instead, we retain the attendance feature. The final dataset resulted in a total of 92 survey items and 131 features after one-hot encoding. In Appendix A, Table A1 and Table A2 outline the 2 domains and 10 constructs, and their respective 131 features. Below, we define each construct and summarize its relevance to adolescent smartphone overdependence.

**Intent to Seek Smartphone Education.** This construct captures adolescents’ intention to engage in preventive education related to smartphone use [48]. Intention to seek smartphone education can be understood as a proxy within the broader help-seeking framework for behavioral addictions. Treatment- or help-seeking studies highlight its association with smartphone struggles across countries [48,49,50] and in other smartphone-accessible issues such as internet gaming disorder, internet addiction, and social network sites [51,52].**Preventive Education.** This reflects actual participation in any educational or intervention programs to prevent smartphone addiction [53,54]. Although evidence for their efficacy and effectiveness remains mixed [55,56,57,58], adolescents who have received such education often demonstrate greater awareness and are more likely to engage in help-seeking behaviors. Those who have received such education tend to have greater awareness of the risks, perceive the programs as more helpful when at high risk, and even demonstrate reduced overdependence and improved self-control in some cases [53,59,60].**Smartphone Use Cases.** This measures the frequency of engagement with different smartphone functions (e.g., gaming, video streaming, and navigation) [61]. Prior studies have found important associations between the types of smartphone use and overdependence across populations, including adolescents [34,62]. Entertainment and social networking applications, in particular, have been linked to increased overdependence due to their reward structures and habit-forming properties [63].**Home Environment.** The home environment includes both physical and psychological aspects of the household context [64]. Prior studies have shown that conditions such as socioeconomic status, family structure, and parental support can shape adolescent behavior and potentially influence smartphone use [65,66]. For instance, socioeconomic status is linked to adolescents’ screen media use [67].**Parental Prevention Efforts.** Parental prevention efforts refer to the strategies parents use to manage and guide their children’s media use, including rule-setting, supervision, and open dialogue [68,69,70]. Active parental prevention efforts have been found to reduce risk of smartphone addiction, while overly restrictive or absent parenting may increase it [71,72,73]. In addition, parental mediation and psychological control significantly predict problematic smartphone use through mechanisms such as psychological reactance and resilience [71,74,75].**Social Support.** Social support refers to the perception that one is cared for, valued, and part of a network of mutual obligations [76]. Strong social support has been associated with reduced smartphone overdependence, potentially by mitigating negative emotions and enhancing self-regulation [77].**Self-Regulation Assessments.** Self-regulation is the ability to control emotions, behaviors, and cognition in pursuit of long-term goals [78,79,80]. A growing body of research suggests that higher self-regulation is associated with lower levels of smartphone overdependence in adolescents [81,82].**Perceived Digital Competence and Risk.** This construct refers to individuals’ perceived abilities to effectively navigate digital environments (e.g., digital content creation and privacy awareness) as well as their perceived smartphone issues (e.g., excessive use and difficulty controlling short-form video consumption). These perceptions may shape how individuals engage with digital technology and manage potential overdependence [83]. Notably, from a cognitive–behavioral perspective, such self-evaluations can reflect cognitive distortions [84]. In the smartphone literature, maladaptive cognitions and impaired self-regulation reinforce problematic use [46,47]. In addition, Perceived Digital Competence has emerged as a relevant construct where prior research has demonstrated that higher digital literacy can buffer risks in some groups but also exacerbate overdependence when combined with stress or limited coping resources [85].**Life Satisfaction.** Life satisfaction refers to a person’s evaluation of their overall well-being and quality of life [86,87,88]. Studies have shown that higher levels of smartphone addiction are correlated with lower life satisfaction among adolescents [82,89].**Smartphone Consequences and Dependence.** Smartphone Consequences and Dependence is characterized by emotional suffering, typically involving symptoms of depression and anxiety [19,90,91]. Excessive smartphone use has been linked to increased psychological distress and related cognitive and neurological effects [19,92].

### 2.4. Artificial Intelligence/Machine Learning Models

This section describes our conceptual framework and development of AI/ML models and then presents our evaluation methods. All analyses were conducted using Python version 3.10.13, with commonly used data science and machine learning libraries (e.g., scikit-learn, pandas, XGBoost, and LightGBM). The codebase can be made available upon reasonable request to the corresponding author.

#### 2.4.1. Conceptual Framework

We build various AI/ML models using two primary approaches: (1) nested modeling and (2) construct-based modeling. In the nested modeling, we begin with simple demographics and add sets of features one construct at a time. The sequence of nesting is based on the risk of psychological discomfort. In other words, we consider the ethical considerations with implementation-informed design. This conceptual approach was chosen to develop the low-risk screener. This would also identify the screener with the minimum features needed for high prediction accuracy. Next, we take a construct-based modeling approach to identify which construct feature sets have the most predictive power and examine the feature importance within. This provides a conceptual approach toward the construct exploration of contributing factors leveraging AI/ML.

#### 2.4.2. AI/ML Classification Modeling

To examine smartphone overdependency among adolescents, we developed classification models to predict our binary outcome regarding any risk of adolescent smartphone overdependency. Here, we discuss our data processing and AI/ML procedures.

##### Data Processing

Our dataset comprises numerical and categorical variables. First, we one-hot encode the categorical variables in our dataset. To standardize our data, we then train a scalar on our training set using the StandardScalar function from the scikit-learn library [93]. This resulting scalar is then applied to the test set to prevent data leakage.

##### AI/ML Procedures

We applied widely used high-performing algorithms to maximize classification performance. They include logistic regression (LogReg), decision trees (DTs), random forest (RF), extreme gradient boosting (XGBoost), multi-layer perceptron (MLP), and light gradient-boosting machine (LightGBM) [94,95,96,97,98]. We then split the data into 80/20 training and out-of-sample test sets. The training set had 1498 adolescents, and the test set had 375 adolescents, which were held constant across algorithms and all experiments. We use 5-fold cross-validation, nested within the training set, for hyperparameter tuning and out-of-sample testing for final performance evaluation. We tune hyperparameters specific to each algorithm using Optuna (an automatic hyperparameter optimization framework) with 50 iterations to minimize mean logarithmic loss across the 5-fold cross-validation (mean val loss) [99]. For MLP and gradient boosting algorithms, we utilize early stopping to prevent overfitting. We report the search spaces and hyperparameters for our reported models in Appendix B. In the final testing, model performance was assessed using three metrics: mean val loss, test loss, and the out-of-sample area under the receiver operating characteristic curve (AUC). The consistency between mean val loss and test loss was used to assess model generalizability, while the out-of-sample AUC was employed to evaluate the model’s ability to discriminate between classes on unseen data [100,101].

#### 2.4.3. Explainable AI (XAI) Techniques

We apply XAI techniques to improve the interpretability of the model by identifying features that meaningfully contribute to its predictions [102,103]. This ensures transparency and interpretability of our classification models for predicting adolescent smartphone overdependency. We use two XAI tools, Shapley Additive exPlanations (SHAP) and partial dependence plots (PDPs) [104,105].

SHAP assess both global (mean feature impact) and local (instance-specific) contributions of each predictor, clarifying how features influence predictions [105]. Additionally, we employed PDPs to analyze key nonlinearities, inflection points, and interaction thresholds in predictor–outcome relationships [106]. Section 3 presents SHAP beeswarm plots and PDPs to illustrate SHAP value distributions and the functional effects of high-impact predictors. We compute SHAP values to quantify and examine the rank and directionality of features, providing insights into how predictors influence predictions. We generate PDPs to visualize global marginal effects on the average prediction, providing insights into nonlinear relationships and slope inflection points.

#### 2.4.4. Place-Stratified Predictive Analysis

We conduct a place-stratified predictive analysis by evaluating the performance of our AI/ML models on sample subsets stratified by the South Korean government administrative urbanicity classifications [41]: (1) metropolitan city, (2) medium or small city, and (3) town or rural district. Group comparisons found 31 significant differences by urbanicity out of 128 one-hot-encoded features (Table 1). This may lead to nationally trained models missing key place-based nuances or inconsistent performance. Using our out-of-sample test set and stratifying the predictions by urbanicity, we recalculate the AUCs, SHAP, and PDPs only for those sample subsets and examine differences in feature importance and trends (see Table A4, Table A5 and Table A6 in Appendix C for group comparisons across all study variables). We use these insights to provide place-based policy profiles in Section 4.1.

## 3. Results

The following sections present the sample descriptive results, an overview of the AI/ML results from each modeling approach (nested and construct-based), and further explore four exemplar models.

### 3.1. Sample Descriptives

A total of 1873 adolescents aged 18 years or younger were included in the analysis, with 1498 in the training set and 375 in the test set. No significant group differences between the training and test sets were observed in any of the study variables (see Section C.2
Table A9, Table A10, Table A11, Table A12 and Table A13 for detailed analysis).

Within the total sample, the mean age was 14.34 years (SD = 2.40), and 51.9% of the participants were of female sex. Further, 44.4% of the participants resided in metropolitan cities, 38.1% in medium/small cities, and 17.5% in towns/rural districts. A majority (71.8%) reported that both parents were working, while 7.8% lived in single-parent households, and 0.7% identified as being from multicultural families, i.e., having at least one foreign-born parent. About 21.8% lived in single-family homes, while the remainder resided in apartments or other multi-unit dwellings. The average monthly household income was KRW 4.25 million (SD = 1.12; approximately USD 3100).

### 3.2. Overview of AI/ML Results

The following sections present the AI/ML results from the nested and construct-based modeling frameworks.

#### 3.2.1. Performance Comparison for Nested AI/ML Classifiers

In the nested framework, 12 nested AI/ML classifiers were trained. Overall, the results show generalizability across training and test sets with consistent mean val loss and test loss of each nested classifier. The baseline model (M0 [Demographics]) performed at chance (AUC = 0.51), and M1 (+Urbanicity) modestly improved performance (AUC = 0.57). Adding a single intent to seek the smartphone education item in M2 increased the AUC to 0.63, but further smartphone prevention education-related features in M3 did not yield gains. A substantial improvement emerged in M4 (+Smartphone Use Cases), which added behavioral features related to smartphone use. This model achieved an AUC of 0.81, an 18-point increase, indicating it as a “good test” [100]. We identify the M4 (+Smartphone Use Cases) as our low-risk screener as it offers high performance while using low-burden non-intrusive features and requires no psychological assessments.

Subsequent models added features related to social context (M5–M7), self-regulation (M8), and Perceived Digital Competence and Risk (M9), leading to incremental improvements. The highest performance was observed in M11 (+Consequences and Dependence), which reached an AUC of 0.92 and qualifies as an “excellent test” [100]. However, the inclusion of psychologically sensitive content and longer assessments makes later models less suitable for early-stage or population-level screening for public health surveillance goals. The strength of the M4 (+Smartphone Use Cases) model lies in its balance between predictive power and feasibility (see Table A14 in Appendix D for more details).

#### 3.2.2. Performance Comparison Between Construct-Based AI/ML Classifiers

In our construct-based modeling framework, we train 12 individual construct-specific AI/ML classifiers. Overall, we find stable performance between the mean val loss and the test loss, indicating generalizability. Three constructs—Smartphone Consequences and Dependence (Smart CD), Perceived Digital Competence and Risk (Perceived DCR), and Smartphone Use Cases (Use Case)—achieve high performance, indicating a “good test”, while all the others indicate “poor tests” [100]. Notably, we find that the Parental Prevention Efforts and constructs related to smartphone education result in “poor tests” despite qualitatively being related to smartphone overdependence.

### 3.3. Four Exemplar Models

We present four exemplar models: (1) the M4 (+Smartphone Use Cases) nested model, (2) the Smartphone Consequences and Dependence (Smart CD) construct-based model, (3) the Perceived Digital Competence and Risk (Perceived DCR) construct-based model, and (4) the Smartphone Use Cases (Use Case) construct-based model. For each model, we present the mean val loss, test loss, and AUC (see Table 2 and Figure 1). Next, using XAI techniques, we examine the feature importance across the models (see Figure 2 and PDP plots below). We then present the place-stratified results regarding each model’s performance (see Table 3) and relative SHAP feature importance (see Figure 3).

#### 3.3.1. M4 (+Smartphone Use Cases) Nested Model

Of the 12 nested classifiers, the M4 (+Smartphone Use Cases) model is the best candidate for a low-risk screener with detection and public health surveillance goals in mind. This model is parsimonious, using 59/131 features, and balances the tradeoff between high performance (AUC = 0.81; Figure 1) and minimizing psychological discomfort. The majority of the predictive power comes from the adolescents’ behavioral use cases of their smartphone requiring no psychological reflection, assessment, or discomfort. These features added a substantial improvement of 18 percentage points, from an AUC of 0.63 in M3 to 0.81 in M4, achieving a “good test” designation [100]. In addition, this model shows good generalizability across training and test sets with stable mean val loss (0.58) and test loss (0.55) scores.

##### Feature Importance

The SHAP beeswarm plot for M4 (+Smartphone Use Cases) is presented in Figure 2A. Overall, higher endorsement of these features predicted a higher likelihood of being at risk of smartphone overdependency. Some notable nuances are that “Edu Intent” (3-point Likert) was the most important and appears to have a thresholding effect, where only high endorsement (red) led to a higher likelihood of smartphone overdependency. In the PDP, this results in only the endorsement of 3 being associated with a likelihood of overdependency (Figure 4A). In contrast, the “Gaming” (8-point Likert) and “Short-Form Video Usage Portion” (4-point Likert) features indicate a clear positive relationship, where higher endorsement (red) is associated with a higher likelihood of overdependency. The PDPs for both features show a linear trend (Figure 4B,C). The PDPs also reveal a notable nonlinear trend in the marginal effect of “Phone Use Case—Navigation” (7-point Likert), where extreme low (0–1) and high (4–6) levels of endorsement led to higher likelihood of overdependency while only medium (2–3) levels of endorsement led to lower likelihood (Figure 4D). This may be due to the integral use of smartphones for navigation, where not using them at all is out of the norm and may indicate overuse of other use cases, while overuse would increase reliance and dependency on one’s smartphone (see Appendix E for all the other PDPs).

##### Place-Stratified Results

The place-stratified analyses revealed that metropolitan city adolescents have the highest performance, with an AUC of 0.87 (compared to 0.81 in the national performance; Table 3). In contrast, the performance among medium/small city adolescents is the lowest, with an AUC of 0.75, which would only indicate a “fair test” [100]. Among town/rural district adolescents, the discriminative power remains about the same, with an AUC of 0.80.

##### Placed-Stratified Feature Importance

The place-stratified analysis of SHAP feature importance found that, across all the strata, “Edu Intent—Intent to Participate in Prevention Program”, “Phone Use Case—Gaming”, and “Phone Use Case Video—Short-Form consistently ranked as the top three predictors of smartphone overdependency. “Phone Use Case—Navigation” ranked moderately across all the regions, appearing between seventh and eighth in all the strata, with no major fluctuations in relative importance. Additionally, “Phone Use Case—Online Learning” ranked higher in metropolitan cities (seventh) compared to the national and town/rural district strata (both ninth) and was the lowest in medium/small cities (tenth). In town/rural districts, “Phone Use Case—Webtoon” was ranked fourth—one position higher than in the other regions—exceeding “Phone Use Case—Selling Goods and Services”, which ranked fifth. Overall, the place-stratified results show strong consistency in the top-ranked predictors, with minor differences in mid- and lower-ranked features by regional context.

#### 3.3.2. Smartphone Consequences and Dependence Classifier Results

The Smart CD model had the highest performance across all the construct-based models, with an AUC of 0.89. However, this construct asks adolescents to complete the highest degree of intrusive reflection specific to their potential problems and dependency issues related to their smartphone use. Therefore, it is not surprising that it is the most predictive, but it also carries the largest potential for psychological discomfort.

##### Feature Importance

The SHAP beeswarm plot for the Smart CD model is presented in Figure 2B. We find that consequences are more predictive than dependence on smartphones for “Stress Relief”. The PDPs revealed three patterns of marginal effects: (1) linear step-wise increase, (2) thresholding effect, and (3) binary effect. “Social Isolation” (3-point Likert) followed the linear, step-wise increase. “Physical Effects” (4-point Likert), “Depression” (4-point Likert), and “Periodic Check Nervousness” (4-point Likert) followed the thresholding effect, where there was an initial increase in predicted probability of overdependence that later flattens out. “Stress Relief” (4-point Likert), “Low Battery Nervousness” (4-point Likert), and “Sleep Issues” (4-point Likert) followed the binary effect. Figure 5 shows examples of each pattern (See Appendix F for all other PDPs).

##### Place-Stratified Results

The Smart CD model performance between the national and stratified-by-urbanicity samples is presented in Table 3. The results show that the performance among metropolitan city adolescents is the highest, with an AUC of 0.91, indicating an “excellent test” [100]. The performance among medium/small city and town/rural district adolescents is the same, with an AUC of 0.87, performing slightly worse than the national model but still in the “good test” range [100].

##### Placed-Stratified Feature Importance

The place-stratified analysis of SHAP feature importance found that, across all the strata—national, metropolitan city, medium/small city, and town/rural district—“Low-Battery Nervousness”, “Physical Effects”, and “Periodic Check Nervousness” consistently appeared as the top contributors to the predicted probability of smartphone overdependence. In the metropolitan city stratum, “Physical Effects” was the most important feature, followed by “Low-Battery Nervousness” and “Periodic Check Nervousness”. This ordering differs from the overall national model, where “Physical Effects” ranked slightly lower. In the medium/small city stratum, the top-ranked features mirrored those of the national sample, with “Low-Battery Nervousness” and “Physical Effects” leading in importance. In the town/rural district stratum, “Low-Battery Nervousness” and “Physical Effects” again ranked highest. Notably, “Sleep Issues” appeared slightly more prominently in this group than in the others. Across all the regions, all the Smart CD features contributed positively to predicted overdependence when endorsed at higher levels. The feature rank order varied only slightly between strata, indicating overall stability in the predictor structure across places.

#### 3.3.3. Perceived Digital Competence and Risk Classifier Results

Next, the Perceived DCR model achieved an AUC of 0.84, asking questions regarding one’s self-perceptions rather than experienced harms.

##### Feature Importance

The SHAP analysis revealed mixed results, where higher endorsement (red) of perceived phone issues was often associated with a higher likelihood of overdependency, and higher endorsement of perceived ability was often protective (Figure 2C). Several deviant cases are discussed. “Perceived Phone Issues—Short-Form Video Usage Control Difficulty” (4-point Likert) revealed that moderate endorsement was more protective than low endorsement. This may reflect instances of denial in use control difficulties. “Perceived Ability—Assess Online Information Reliability” and “Perceived Ability—Online Privacy Awareness” were not protective, and higher endorsement (red) predicted higher likelihood of overdependency, perhaps reflecting illusions of control and cognitive distortions. “Perceived Ability—Online Social Issue Engagement” and “Perceived Ability—Digital Content Creation” were also not protective and may reflect a higher reliance on smartphones for socialization and creativity. Two notable parabolic or higher-order trends emerged for the Perceived DCR model PDPs in their marginal effects: (1) a negative to positive curve, and (2) a positive to negative to positive curve, revealing complex cognitive patterns in smartphone overdependency. Figure 6 shows examples of each pattern (see Appendix G for all the other PDPs).

##### Place-Stratified Results

The performance of the Perceived DCR model stratified by urbanicity is presented in Table 3. Across all the strata, the performance remains in the “good test” range. Among metropolitan city adolescents, the AUC is 0.85, while, among both the medium/small city and town/rural district adolescents, the AUC is 0.83.

##### Placed-Stratified Feature Importance

Across all the strata, three perceived phone issues consistently ranked highest: “Smartphone Overdependence”, “Excessive Smartphone Using Time”, and “Short-Form Video Usage Control Difficulty”. These self-perceived issues demonstrated stable predictive strength across metropolitan and medium/small cities. In contrast, the town/rural district stratum showed slight reordering in feature importance. While “Smartphone Overdependence” and “Excessive Smartphone Using Time” remained top predictors, “Negative side effect on gaming” ranked higher than“Short-Form Video Usage Control Difficulty”, and “Perceived Ability—Online Social Issue Engagement” appeared more prominently. Specifically, “Online Social Issue Engagement” ranked fifth in the towns/rural districts stratum (vs. seventh in the others), and “Negative Side Effects on Messenger” ranked sixth (vs. tenth in the others). Overall, while the top self-perceived dependence and usage difficulty features remained stable across regions, the town/rural district model showed relatively greater importance for social and emotional factors—such as issue engagement and messaging-related effects—while cognitive–evaluative skills and digital production activities showed relatively reduced predictive influence.

#### 3.3.4. Smartphone Use Case Results

The final “good test” classifier is the Use Case model with an AUC of 0.80, which only uses behavioral tracking-related indicators. The Use Case model suggests potential in leveraging raw behavioral tracking data directly from smartphones.

##### Feature Importance

The SHAP analysis revealed nuances in which types of Smartphone Use Cases are predictive of overdependency (Figure 2D). Indeed, 13 of the top 20 important use cases had a predominantly positive relationship, where higher endorsement (red) was associated with overdependency. The three most important features of this pattern were “Gaming”, “Short-Form Video Usage Portion”, and “Webtoon”. Four of the top twenty features had a predominantly negative relationship, where lower endorsement (blue) was associated with overdependency. The three most important features of this pattern were “Navigation”, “News”, and “Health”. This may reflect a bifurcation where Smartphone Use Cases for entertainment are more likely to predict overdependency, while use cases that capture a non-entertainment utility value are more protective. The Use Case PDPs generally supported the SHAP findings, with minor nuances or thresholding effects across Likert responses, but still held overall positive or negative trends. Notably, the PDP of the “Navigation” (7-point Likert) feature was consistent with the nested M4 (+Smartphone Use Cases) model, where the same trend and cutoffs were found. Extreme low (0–1) and high (4–6) levels of endorsement led to a higher likelihood of overdependency, while only medium (2–3) levels of endorsement led to a lower likelihood. This consistency across the models supports the robustness of this feature’s relationship. Figure 7 shows examples and presents the “Navigation” PDPs (see Appendix H for all the other PDPs).

##### Place-Stratified Results

The performance of the Use Case model stratified by urbanicity is presented in Table 3. These results show that, among metropolitan city adolescents, there is a substantial jump in performance with an increase of 8 percentage points for an AUC of 0.88. However, the performance substantially decreases for the medium/small city adolescents, with a decrease of 10 percentage points for an AUC of 0.70, falling into the “fair test” range [100]. The performance among towns/rural districts decreases marginally to 0.79 but falls from a “good test” to “fair test” [100].

##### Placed-Stratified Feature Importance

Across all the strata, four features consistently emerged as top predictors: “Gaming”, “Short-Form Video Usage Portion”, “Webtoon”, and “Navigation”. “Gaming”, “Short-Form Video Usage Portion”, and “Webtoon” consistently showed strong positive contributions to predicted risk. “Navigation” was the only feature with a mixed effect, consistently ranked top but associated with slightly lower predicted risk at moderate levels of use. In the metropolitan city stratum, “Gaming” ranked first, followed by “Navigation”, “Webtoon”, and “Short-Form Video Usage Portion”. Phone use for “News” showed negative contributions to the predictions in this stratum. Overall, behavioral features with immersive or compulsive qualities dominated. In the medium/small city stratum, “Navigation” and “Short-Form Video Usage Portion” were the top predictors, followed by “Gaming” and “Webtoon”. Notably, “Listening to Music” ranked tenth only in this region, indicating that more passive forms of phone use may be more salient here. In the town/rural district stratum, the top four features mirrored those in the national sample. However, there was greater diversity in the top ten features, including “Photography” (ranked ninth), which did not appear in the other regions. This suggests that expressive and leisure-oriented smartphone behaviors play a more prominent role in rural contexts. Features like “Selling Goods and Services”, “Online Learning”, “Messenger”, “Online Shopping”, and “Scheduling” were present across all the strata with moderate contributions, while “Listening to Music” appeared in all the regions except metropolitan cities, ranking the lowest in predictive strength but showing regional relevance.

## 4. Discussion

This study is the first to develop a high-performing, low-risk AI/ML-based screener for adolescent smartphone overdependence in South Korea using conceptually informed modeling and a place-stratified analysis approach. By combining ethical considerations with a conceptually informed design, we identified a parsimonious screening model that balances predictive power with psychological safety. The M4 (+Smartphone Use Cases) model, leveraging only behavioral use data and demographic characteristics, achieved an AUC of 0.81 nationally and 0.87 in metropolitan cities, comparable to more invasive models using psychological assessments. This confirms our working hypothesis that a minimal set of non-invasive indicators can yield strong predictive power.

These results align with prior studies that emphasized the importance of smartphone use patterns, particularly entertainment-related activities, in predicting overdependence [33,63]. Our XAI analyses provided further insight, showing that use cases like gaming and short-form video were strong predictors of risk, while utilitarian use cases such as navigation exhibited a nonlinear pattern, where both low and high engagement signaled higher risk. This thresholding and parabolic trend in use mirrors earlier findings in digital addiction and behavioral tracking, suggesting that moderate use may reflect healthy regulation, whereas extremes indicate dependency [37].

In parallel, our construct-based models expand on the potential of AI/ML conceptual exploration. Our models confirmed the theoretical relevance of both perception- and consequence-based constructs. The Smartphone Consequences and Dependence model achieved the highest performance (AUC = 0.89), but its predictive strength likely reflects conceptual redundancy with the overdependence outcome itself. Many items in this construct directly assess perceived harm and experienced behavioral symptoms that closely mirror the criteria used to define smartphone overdependence, limiting their theoretical distinctiveness despite high classification accuracy.

In contrast, the Perceived Digital Competence and Risk model (AUC = 0.84) provides more meaningful theoretical insight. It aligns closely with cognitive–behavioral frameworks by examining how adolescents evaluate and perceive their own ability and issues with smartphone use. Features related to perceived control, awareness of usage patterns, and beliefs about digital competencies correspond to key mechanisms in models of behavioral addiction and cognitive distortions. Cognitive distortions are systematic errors in thinking that bias perceptions of self, the world, or one’s control, and they are central mechanisms targeted in cognitive–behavioral therapy [107]. The nonlinear and parabolic trends observed in this model’s PDPs suggest that adolescents’ cognitive self-perceptions are complex, potentially involving both denial and overconfidence, patterns that are well-documented in cognitive–behavioral theory [107,108,109] and often targeted in CBT-based interventions for addictive behaviors [110]. Notably, some indicators, such as high confidence in digital literacy, were associated with increased risk, perhaps reflecting distorted cognitions or rationalizations, well-known patterns in CBT models that justify and maintain excessive use [84,107], which merit further psychological investigation. In the following section, we present place-based policy implication profiles based on our place-stratified analysis.

### 4.1. Place-Based Policy Implication Profiles

We identify tangible implications that can guide place-based policy responses across regional contexts for the prevention and intervention of smartphone overdependence among adolescents. The NISA survey uses the classification of municipalities outlined in South Korea’s Local Autonomy Act to stratify urbanicity. The act distinguishes municipalities by population size and administrative capacity, designating metropolitan cities (≥500,000 residents), medium/small cities (≥50,000 but <500,000), and towns/rural districts as distinct units of local governance [41]. This stratification is also aligned with high–middle–low accessibility clusters in South Korea’s social infrastructures, illustrating how access patterns shape opportunity structures [111]. Each distinct stratum experiences its own unique exposures and consequences, requiring tailored preventive strategies. This approach is consistent with the prior literature on spatial inequalities. For instance, Parker et al. [112] showed that urban, suburban, and rural communities differ in lived experiences in ways that extend beyond a simple binary, underscoring the need for nuanced stratification. By linking smartphone overdependence to broader discussions about urbanicity differences, our place-based profiles extend existing policy discussions toward more context-sensitive responses that can be readily utilized by the South Korean government.

#### 4.1.1. Metropolitan Cities

Metropolitan cities in Korea are densely populated, often with several million residents, and heavily reliant on mass transit systems [18]. Adolescents in these areas face intense academic pressure and constant digital connectivity, with evidence showing stronger associations between overdependence and anxiety in metropolitan youth [18]. Within this context, our result shows that metropolitan adolescents frequently coincide with comparatively higher perceived digital competence in information for school/work and utility-related use cases (e.g., scheduling, Zoom meetings, investment, and educational web search) (Table 1), while the physical consequences of smartphone use and engagement in online learning, gaming, and webtoons stood out as salient predictors of overdependence compared to other strata (Figure 3). Of note, gaming was a particularly salient predictor of overdependence among metropolitan adolescents, ranking within the top 2, while other strata ranked gaming lower (Figure 3). Model performances were strongest in this stratum (average AUC = 0.88; Table 3). This aligns with Sapienza et al. [37], who found that urban residents worldwide favor utility and multifunctional applications to navigate complex city environments. Policy approaches in metropolitan areas should therefore emphasize media literacy components focused on self-monitoring and balanced digital engagement, rather than broad usage restrictions. School-based initiatives can effectively incorporate media literacy components focused on self-monitoring techniques and healthy smartphone usage discussions among peers, with specific attention to gaming and the negative physical effects of phone use. Additionally, given the strong association between anxiety-driven behaviors and smartphone overdependence, which has been particularly shown in high-density urban settings [38], integrating targeted mental health services within schools is recommended to directly address compulsive patterns of smartphone use.

#### 4.1.2. Medium and Small Cities

Medium and small cities in Korea typically range from several hundred thousand to just over one million residents, with mixed public and private transportation and stronger community-based schooling [33]. Adolescents here are exposed to moderate digital intensity and often benefit from closer community ties, with municipal youth centers and local after-school programs providing key venues for prevention [33]. In our results, model fit was lowest in this stratum across models (average AUC = 0.79; Table 3), with Table 1 indicating the highest levels of *low battery nervousness* and greater TikTok and watching videos engagement, alongside the lowest social support across all sub-indicators. This reflects patterns noted in China, where rural-to-urban transitional contexts were associated with smartphone use as a coping tool for loneliness and anxiety [39,40]. In Figure 3, short-form video usage was consistently a more important predictor than gaming, a contrast to metropolitan adolescents, suggesting differing approaches to coping. Policies here should deliver tailored education that normalizes healthy stress management beyond digital devices, such as emotional literacy workshops, digital citizenship training, and habit-tracking supports. Interventions that highlight the risks of entertainment-driven coping while offering practical alternatives may reduce compulsive reliance in these middle-tier contexts.

#### 4.1.3. Towns and Rural Districts

Rural districts are sparsely populated, with limited broadband infrastructure and fewer specialized health services [113]. Adolescents in towns and rural districts in the present study predominantly reported smartphone use for leisure-related activities (e.g., video streaming, listening to music, and short-form YouTube) and the highest perceived digital competence in content creation (Table 1), though gaming was consistently less important in predicting overdependence relative to other strata (Figure 3). These patterns are consistent with Sapienza et al. [37], who noted that rural users disproportionately use smartphones for entertainment, likely due to limited offline recreation opportunities. Similar rural-specific vulnerabilities have been documented in Bangladesh, where varying degrees of urbanicity predicted smartphone addiction in children, and in China, where rural adolescents reported greater dependence due to isolation and anxiety [39,40]. South Korean data further suggest that overdependence in rural adolescents is associated with anxiety and psychosocial difficulties [18]. In addition, Figure 3 shows that relative to other strata, perceived competence in online social issue engagement and messenger use stood out as more salient predictors of overdependence. These findings underscore that rural smartphone use often reflects structural and social constraints rather than individual preference. Model performances were mid-range (average AUC = 0.82; Table 3), with sleep issue consequences being a relatively more salient predictor (Figure 3). Accordingly, rural policies must go beyond limiting screen time to address structural and social constraints, such as expanding offline youth facilities, strengthening community-driven recreation, and establishing safe online interaction guidelines. Guidelines should address consequences such as sleep issues and discuss local engagement in social issues with peers. As Song [113] suggested, telehealth services should also be prioritized to compensate for the lack of in-person resources, alongside school-linked screening and parent/teacher training to bridge prevention gaps.

### 4.2. Limitations and Future Research

This study has several limitations. First, all the data were derived from self-report surveys, including behavioral indicators. Future research should explore the integration of passive smartphone tracking data, which may offer higher ecological validity and remove social desirability bias. In addition, Ecological Momentary Assessment methods can improve these studies. Second, while we stratified the models by urbanicity, other contextual variables—such as school type or family environment—could further refine prediction. Third, while our models were interpretable using SHAP and PDP, further work using longitudinal data is needed to establish causal pathways.

Future studies should also investigate protective features and resilience factors, particularly in adolescents who report high usage but do not develop dependency. Additionally, given the complex nonlinear relationships found in Perceived Digital Competence and Risk indicators, qualitative or mixed-methods research may help to unpack underlying cognitive distortions of smartphone use.

Although biological sex was accounted for as a demographic variable, sex-disaggregated analyses were not conducted in this study. Prior research in Korean adolescents has produced mixed findings regarding sex differences in smartphone overdependence, and no definitive pattern has been established. For example, Noh et al. [8] reported a higher risk among girls, but other studies did not emphasize or systematically examine sex differences [114]. Moreover, Chen et al. [115] found no significant sex differences in smartphone addiction prevalence among Chinese medical college students. Given this lack of consistent evidence and considering the study’s primary objective of developing an overall predictive model, subgroup analyses were not pursued. Future research should explore sex variations more directly and robustly.

## 5. Conclusions

This study’s contributions are both methodological and practical. Methodologically, this study demonstrates the utility of conceptually informed construct modeling over “kitchen sink” approaches. Psychologically salient constructs—particularly those relating to use patterns, Perceived Digital Competence and Risk, and Consequences and Dependence—outperform broader demographic or contextual predictors. Practically, our work supports the adoption of low-burden adolescent-friendly screeners in school settings, aligning with ethical frameworks (e.g., the Belmont Report; APA Code of Ethics) that call for minimal harm in youth research. Conceptually, this study illustrates the use of AI/ML to uncover potential theoretical phenomena. In our case, the cognitive patterns point to potential cognitive distortions. Policy-wise, our work contributes to the growing field of place-based AI/ML by demonstrating that predictive accuracy and feature importance vary by urbanicity. Our results suggest that the risk of overdependence is not uniform across regions. Local social, technological, and cultural contexts shape which factors matter most and how to address them. These findings support the development of regionally tailored public health messaging and digital wellness interventions. Adolescents from metropolitan cities can benefit the most from interventions that target gaming and healthy use of utilitarian functions while emphasizing the risk for negative physical effects of smartphone overdependence. Adolescents from medium or small cities consistently showed lower model performance across the classifiers, suggesting that these communities may exhibit more heterogeneous or undercaptured patterns of smartphone use. In addition, these cities can utilize their existing community resources to increase outreach and intervention. Lastly, adolescents at risk of overdependence from towns and rural districts appear to be longing for community, as suggested in their higher associations with messenger use and perceived competence in online social issue engagement. Recommended interventions involve the empowerment of youth to develop communities and local engagement in social issues on the ground. These nuances emphasize the need for locally adaptive tools and support critiques that national models risk obscuring regional variation [116].

## Figures and Tables

**Figure 1 ijerph-22-01515-f001:**
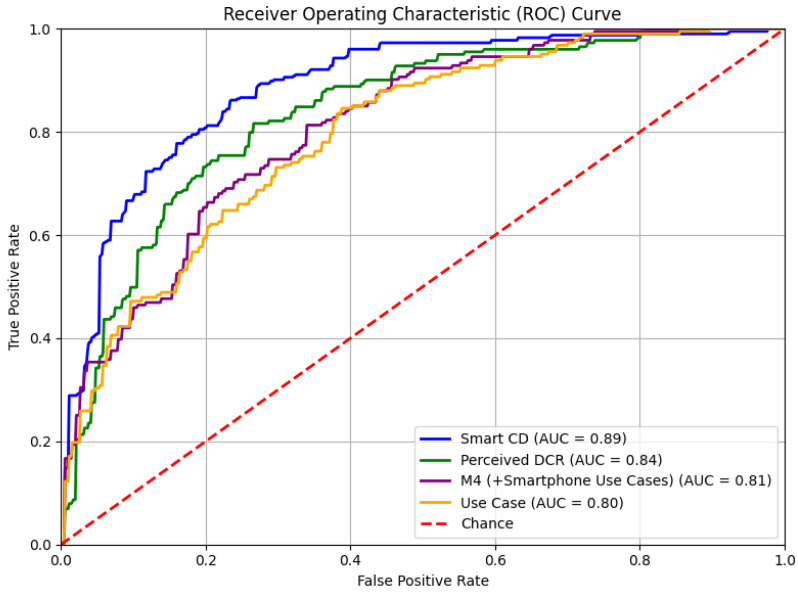
The area under the receiver operating characteristic curves (AUCs) of the M4 (+Smartphone Use Cases) nested model, the Smartphone Consequences and Dependence (Smart CD) construct-based model, the Perceived Digital Competence and Risk (Perceived DCR) construct-based model, and the Smartphone Use Cases (Use Case) construct-based model are presented.

**Figure 2 ijerph-22-01515-f002:**
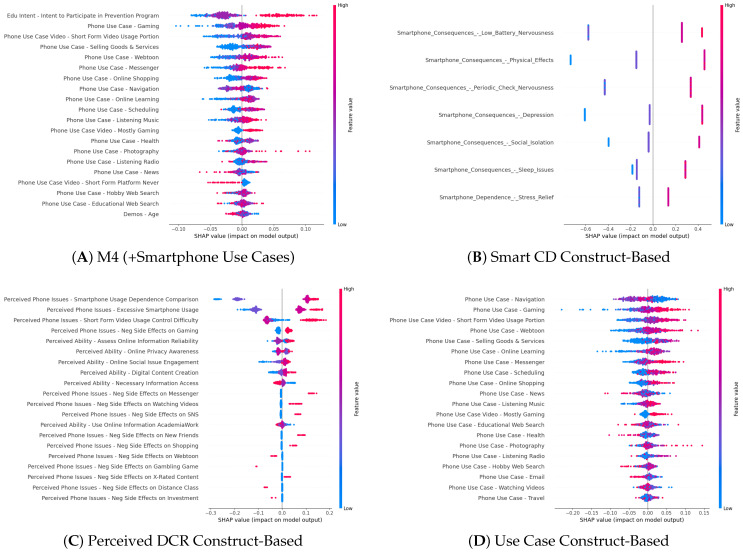
The SHAP beeswarm plots of the M4 (+Smartphone Use Cases) nested model, the Smartphone Consequences and Dependence (Smart CD) construct-based model, the Perceived Digital Competence and Risk (Perceived DCR) construct-based model, and the Smartphone Use Cases (Use Case) construct-based model are presented.

**Figure 3 ijerph-22-01515-f003:**
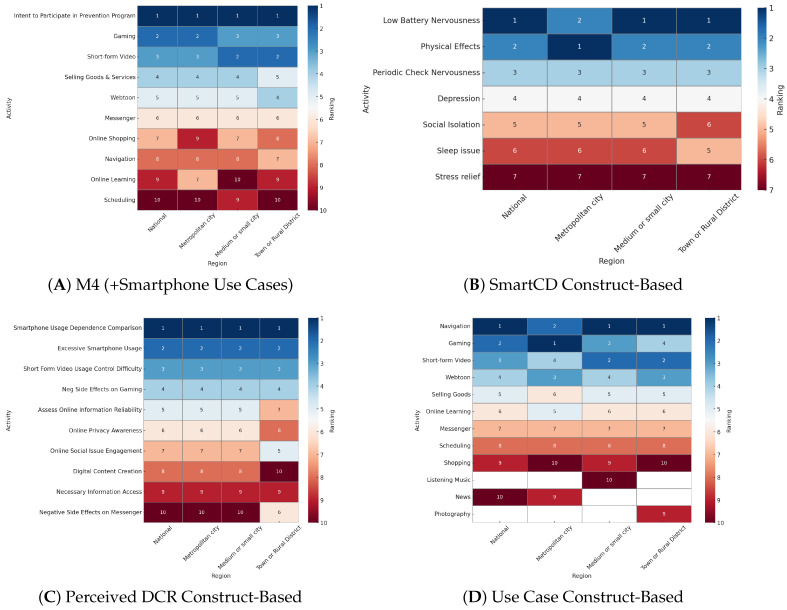
The place-stratified SHAP feature importance rankings of the M4 (+Smartphone Use Cases) nested model, the Smartphone Consequences and Dependence (Smart CD) construct-based model, the Perceived Digital Competence and Risk (Perceived DCR) construct-based model, and the Smartphone Use Cases (Use Case) construct-based model are presented.

**Figure 4 ijerph-22-01515-f004:**
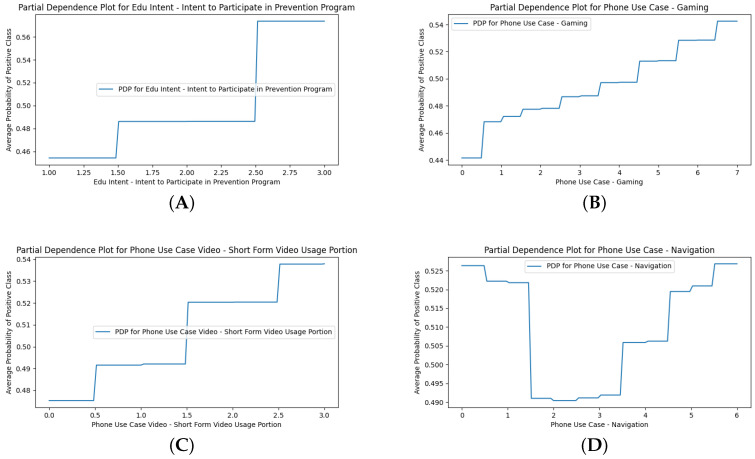
Partial dependence plots (PDPs) of select features in the nested M4 (+Smartphone Use Cases) model. (**A**) shows the PDP for “Edu Intent—Intent to Participate in Prevention Program”. (**B**) shows the PDP for “Phone Use Case—Gaming”. (**C**) shows the PDP for “Phone Use Case Video—Short-Form Video Usage Portion”. (**D**) shows the PDP for “Phone Use Case—Navigation”.

**Figure 5 ijerph-22-01515-f005:**
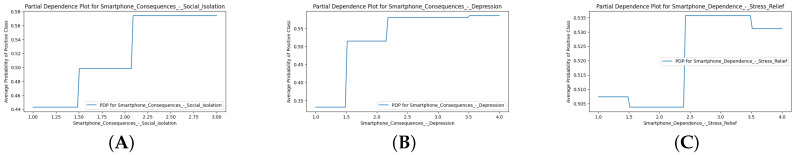
PDP examples of the three marginal effect patterns from the Smart CD model. (**A**) shows “Social Isolation” as the linear step-wise increase pattern. (**B**) shows the “Depression” as the thresholding effect pattern. (**C**) shows the “Stress Relief” as the binary effect.

**Figure 6 ijerph-22-01515-f006:**
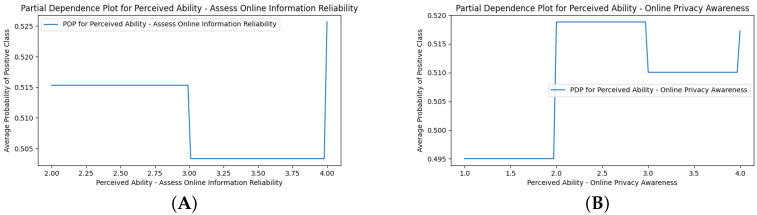
PDP examples of the three marginal effect patterns from the Perceived DCR model. (**A**) shows the “Perceived Ability—Assess Online Information Reliability” as the negative to positive curve. (**B**) shows the “Perceived Ability—Online Privacy Awareness” as the positive to negative to positive curve.

**Figure 7 ijerph-22-01515-f007:**
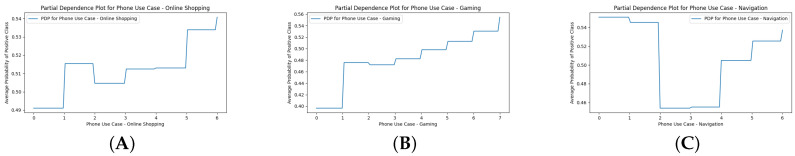
PDP examples and of the “Navigation” feature from the Use Case model. (**A**) shows “Phone Use Case—Online Shopping” with several dips but overall retaining a positive relationship. (**B**) shows the “Phone Use Case—Gaming” with an initial thresholding jump but overall retaining a positive relationship. (**C**) shows the “Phone Use Case—Navigation” with a parabolic curve.

**Table 1 ijerph-22-01515-t001:** Significantly different place-stratified features by urbanicity (N = 1873).

Construct	Variable	Metropolitan City	Medium or	Town or
			Small City	Rural District
		(n = 831)	(n = 714)	(n = 328)
Home Environment	House (n (%)) *	167 (20.1%)	152 (21.3%)	90 (27.4%)
Smartphone Use Cases	Messenger (M (SD)) ***	5.31 (1.46)	5.41 (1.42)	5.02 (1.75)
	Scheduling (M (SD)) **	2.06 (2.13)	1.86 (2.06)	1.62 (2.08)
	Health (M (SD)) *	1.76 (2.04)	1.59 (1.91)	1.46 (1.98)
	Zoom Meetings (M (SD)) *	0.73 (1.57)	0.54 (1.33)	0.59 (1.41)
	Investment (M (SD)) ***	0.86 (1.52)	0.63 (1.28)	0.62 (1.29)
	Watching Videos (M (SD)) *	5.25 (1.62)	5.32 (1.58)	5.02 (1.81)
	Listening Music (M (SD)) ***	4.51 (1.97)	4.48 (2.04)	4.03 (2.35)
	Adult Content (M (SD)) ***	0.50 (1.15)	0.33 (0.94)	0.30 (0.94)
	Gambling (M (SD)) ***	0.48 (1.20)	0.29 (0.87)	0.32 (1.04)
	News (M (SD)) **	2.57 (2.21)	2.30 (2.04)	2.16 (2.16)
	Educational Web Search (M (SD)) **	4.06 (1.81)	3.84 (1.78)	3.64 (2.26)
	Navigation (M (SD)) *	2.42 (1.92)	2.37 (1.95)	2.12 (2.00)
	Short-Form YouTube (n (%)) **	443 (53.3%)	315 (44.1%)	162 (49.4%)
	TikTok (n (%)) *	145 (17.4%)	165 (23.1%)	68 (20.7%)
	Mostly Food (n (%)) *	130 (15.6%)	80 (11.2%)	47 (14.3%)
Perceived Digital Competence and Risk	Gaming Side Effects (n (%)) **	353 (42.5%)	313 (43.8%)	108 (32.9%)
	Issue Engagement (M (SD)) *	2.80 (0.85)	2.70 (0.85)	2.81 (0.81)
	Content Creation (M (SD)) *	2.49 (0.98)	2.43 (0.98)	2.62 (0.99)
	Privacy Awareness (M (SD)) ***	2.66 (0.88)	2.57 (0.86)	2.79 (0.86)
	Info for School/Work (M (SD)) ***	2.89 (0.87)	2.67 (0.94)	2.81 (0.92)
Consequences and Dependence	Low-Battery Nervousness (M (SD)) **	2.71 (0.83)	2.85 (0.86)	2.75 (0.85)
Parental Prevention	Teaches Use (n (%)) **	397 (47.8%)	284 (39.8%)	130 (39.6%)
	Teaches Pros/Cons (n (%)) *	514 (61.9%)	422 (59.1%)	176 (53.7%)
	Recommends Apps (n (%)) ***	396 (47.7%)	277 (38.8%)	152 (46.3%)
Social Support	Family Support (M (SD)) **	3.29 (0.69)	3.17 (0.75)	3.26 (0.66)
	Friend Support (M (SD)) *	3.13 (0.67)	3.07 (0.69)	3.17 (0.68)
	Fairness Perception (M (SD)) ***	2.93 (0.71)	2.81 (0.75)	3.03 (0.79)
Self-Regulation	Digital Detox (n (%)) **	210 (25.3%)	153 (21.4%)	55 (16.8%)
	Designated Placement (n (%)) *	286 (34.4%)	200 (28.0%)	105 (32.0%)
	Work Mode (n (%)) *	414 (49.8%)	316 (44.3%)	136 (41.5%)

Note: Values are means (SD) or counts (percentage). One-way ANOVA used for continuous; chi-square for categorical. Significance: * *p* < 0.05, ** *p* < 0.01, and *** *p* < 0.001.

**Table 2 ijerph-22-01515-t002:** Performance metrics across the four exemplar AI/ML models for smartphone overdependence.

Model Name/Construct	Shorthand	Algorithm	AUC	Mean Val Loss	Test Loss	Num Features
+Smartphone Use Cases	M4	RF	0.81	0.58	0.55	59
Smartphone Consequences and Dependence	Smart CD	LightGBM	0.89	0.50	0.42	7
Perceived Digital Competence and Risk	Perceived DCR	XGB	0.84	0.52	0.49	34
Smartphone Use Cases	Use Case	XGB	0.80	0.59	0.54	48

Note: We compare AUC (area under the receiver operating characteristic curve), mean validation loss, test loss, and number of features across models. Algorithms used are random forest (RF), light gradient-boosting machine (LightGBM), and extreme gradient boosting (XGBoost).

**Table 3 ijerph-22-01515-t003:** AUCs stratified by urbanicity.

Model	National AUC		Place-Stratified AUC	
		Metropolitan City	Medium or Small City	Town or Rural District
M4 (+Smartphone Use Cases)	0.81	0.87	0.75	0.80
Smart CD Construct	0.89	0.91	0.87	0.87
Perceived DCR Construct	0.84	0.85	0.83	0.83
Use Case Construct	0.80	0.88	0.70	0.79

Note: The place-stratified AUCs of the M4 (+Smartphone Use Cases) nested model, the Smartphone Consequences and Dependence (Smart CD) construct-based model, the Perceived Digital Competence and Risk (Perceived DCR) construct-based model, and the Smartphone Use Cases (Use Case) construct-based model are presented.

## Data Availability

The data used in this study were obtained from the Public Data Portal of the Republic of Korea (https://www.data.go.kr/data/15038425/fileData.do, accessed on 20 May 2025). This is a publicly available dataset provided by a government agency and does not require special access permission.

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
