# Peer review of "Adolescent Smartphone Overdependence in South Korea: A Place-Stratified Evaluation of Conceptually Informed AI/ML Modeling"

_ijerph, 2025, doi:10.3390/ijerph22101515_

Round 1

Reviewer 1 Report

Comments and Suggestions for Authors

Dear Editor(s) and Authors,

Thank you for the opportunity to review this manuscript.

The study investigates smartphone overdependence among South Korean adolescents, applying conceptually informed AI/ML to develop a low-risk, high-accuracy screening tool, explore key psychological constructs, and generate region-specific policy insights. Using data from 1,873 adolescents, a 59-feature model achieved strong predictive accuracy (AUC = 81.5%), with smartphone use patterns contributing approximately 20%. Key constructs included Smartphone Use Cases, Perceived Digital Competence & Risk, and Consequences & Dependence. Regional analyses revealed performance variation (AUC = 71.4–91.1%) and local differences. These findings highlight the potential of theory-driven AI/ML for detection, monitoring, and targeted public health policy.

The study is relevant for its topic, methodology, and implications for health policy. Nevertheless, I have several suggestions to improve the manuscript. My comments are organized by section rather than by priority.

TITLE
The title is long (18 words) and does not succinctly capture the main idea. The authors are encouraged to make it more focused and concise. The abstract keywords are comprehensive and could include elements not covered in the title.

ABSTRACT
The abstract provides a thorough overview but could be strengthened by reporting the participants’ mean age, standard deviation, and gender distribution.

INTRODUCTION
The aims of the study are clearly stated in both the abstract and the text. The introduction provides sufficient methodological context but insufficient clarity about the construct under investigation.

  • I recommend defining “smartphone overdependence” at the outset, including its nature, symptoms, and main features, ideally based on the scale used in the study.
  • In Section 1.1, the authors state that the study is grounded in public health and social sciences. However, the introduction is largely methodological and does not adequately review prior literature in these fields. I recommend adding paragraphs that:
    a) summarize recent key findings relevant to the study’s aims;
    b) highlight similarities and differences across these studies;
    c) explain reasons for divergent or contradictory findings;
    d) identify research gaps related to the study’s objectives;
    e) contextualize the current research within the cultural and demographic characteristics of the study population.

METHODS
The methods are rigorous, well described, and appropriate for the aims. The procedures and instruments are clearly outlined, enabling replication. The statistical methods are also suitable. Improvements could include:

  • Line 153: Provide more information on the reliability and validity of the Smartphone Overdependence Scale in the current sample.
  • Line 174–175: Add citations to support the statement: “Higher intention to seek education has been associated with reduced risk of overdependence.”
  • Line 177–180: Also add citations to support: “Although evidence for their effectiveness remains mixed, adolescents who have received such education often demonstrate greater awareness and are more likely to engage in help-seeking behaviors.”

RESULTS
The results are clearly presented, with tables, figures, and the appendix effectively complementing the text.

Regarding gender, please consider the Instructions for Authors on the Journal’s website, which require:

  • description of whether sex and/or gender differences may be expected;
  • reporting how sex and/or gender were accounted for in the study design;
  • provision of disaggregated data by sex and/or gender, where appropriate;
  • discussion of respective results. If a sex/gender analysis was not conducted, a rationale should be provided in the Discussion.

DISCUSSION
The discussion appropriately addresses the findings in relation to previous studies and the study’s aims. Broader public health implications and limitations are noted, and future research directions are outlined. Suggested improvements:

  • Line 550–553:
    1. Clarify the frameworks referenced in the statement: “It aligns closely with cognitive-behavioral frameworks by examining how adolescents evaluate and perceive their own ability and issues with smartphone use” (add specific citations).
    2. Support with citations the claim that features related to perceived control, awareness of usage patterns, and beliefs about digital competencies correspond to mechanisms in models of behavioral addiction and cognitive distortions.

The references are generally adequate, but additional citations following the above suggestions would strengthen the manuscript.

In conclusion, this is a timely study with a well-written manuscript, rigorous methodology, and valuable findings, especially given the novel analytical approach. While I recommend the article for consideration in the International Journal of Environmental Research and Public Health, major revisions are necessary.

I wish the authors success in revising and publishing their work.

Author Response

Comment 1:

TITLE

The title is long (18 words) and does not succinctly capture the main idea. The authors are encouraged to make it more focused and concise. The abstract keywords are comprehensive and could include elements not covered in the title.

Response 1:

We thank the reviewer for the suggestion to make our title more focused and concise while allowing our abstract keywords to cover elements. We have revised the title as follows:

Please refer to the Title.

“Adolescent Smartphone Overdependence in South Korea: A Place-Stratified Evaluation of Conceptually Informed AI/ML Modeling”

Comment 2:

ABSTRACT

The abstract provides a thorough overview but could be strengthened by reporting the participants’ mean age, standard deviation, and gender distribution.

Response 2:

We thank the reviewer for this suggestion and have incorporated it into our abstract.

Please refer to the Abstract.

“Across the sample, adolescents were about 14 years old (SD=2.4) and equally distributed by sex (48.1% male).”

Comment 3:

INTRODUCTION

The aims of the study are clearly stated in both the abstract and the text. The introduction provides sufficient methodological context but insufficient clarity about the construct under investigation.

I recommend defining “smartphone overdependence” at the outset, including its nature, symptoms, and main features, ideally based on the scale used in the study.

Response 3: We thank the reviewer for their attention to the primary construct under investigation and for their suggestion to begin with this in our introduction. We have added a deeper discussion on “smartphone overdependence”, anchoring it to the scale used in the study.

Please refer to the first paragraph on page 2 in the Introduction section.

“This mass exposure and rising engagement have fueled growing concerns about excessive and poorly controlled use. Researchers have conceptualized this phenomenon as a potential behavioral addiction [4, 5], emphasizing impaired self-regulation, functional impairment, and parallels with other behavioral addictions. This aligns with the International Classification of Diseases (ICD)-11, which highlights loss of control and persistence despite adverse consequences as defining features of addictive disorders [6]. In South Korea, government policy formally defines this as smartphone overdependence, a state in which excessive use reduces regulation and produces problematic consequences [7]. Smartphone overdependence is defined as a maladaptive state characterized by salience, impaired self-regulation, and negative social, physical, and academic consequences, and it is measured through the validated Smartphone Overdependence Scale used in Korea’s national surveys [8]. This construct is theoretically and clinically relevant as impulsivity and emotional dysregulation have been identified as core mechanisms [4, 9, 10], while personality characteristics and mental health vulnerabilities further increase risk [11]. Empirical evidence also demonstrates associations with poor sleep recovery and physical inactivity [12], disordered eating behaviors [13], and heightened suicide risk [14]. These findings clarify that smartphone overdependence extends beyond screen time, representing a significant behavioral health concern that warrants policy and clinical attention. The Smartphone Overdependence Scale, widely used in public health monitoring, captures tolerance, withdrawal, and daily life disturbance, reflecting a marked decline in self- control such that smartphone use becomes the most salient activity in daily life [15].”

Comment 4:

In Section 1.1, the authors state that the study is grounded in public health and social sciences. However, the introduction is largely methodological and does not adequately review prior literature in these fields. I recommend adding paragraphs that:

a) summarize recent key findings relevant to the study’s aims;

b) highlight similarities and differences across these studies;

c) explain reasons for divergent or contradictory findings;

d) identify research gaps related to the study’s objectives;

e) contextualize the current research within the cultural and demographic characteristics of the study population.

Response 4: We thank the reviewer for their structured suggestion on the introduction. We have integrated this structure into our manuscript, adding additional public health and social science literature.

Please refer to the Introduction section on pages 2-3.

“(page 2, first paragraph) Smartphones have become deeply embedded in modern life, particularly for adolescents who rely on them for education, entertainment, and social interaction. In South Korea, nearly all adolescents (99.1%) owned a smartphone in 2024 [1], and screen time has continued to rise at an estimated annual growth rate of 12.6% since 2016 [2]. By 2021, adolescents reported approximately five hours of daily smartphone use [ 3].

(page 3, last paragraph of introduction) In South Korea, where nearly all adolescents own smartphones and national policy formally recognizes overdependence, the development of such models is both urgent and essential. The present study directly responds to this need by integrating conceptual frameworks with AI/ML to create adolescent-friendly, ethically minimal, and place- sensitive approaches to detection and policy. The specific aims and contributions are outlined in the following section.”

Comment 5:

METHODS

The methods are rigorous, well described, and appropriate for the aims. The procedures and instruments are clearly outlined, enabling replication. The statistical methods are also suitable. Improvements could include:

  • Line 153: Provide more information on the reliability and validity of the Smartphone Overdependence Scale in the current sample.
  • Line 174–175: Add citations to support the statement: “Higher intention to seek education has been associated with reduced risk of overdependence.”
  • Line 177–180: Also add citations to support: “Although evidence for their effectiveness remains mixed, adolescents who have received such education often demonstrate greater awareness and are more likely to engage in help-seeking behaviors.”

Response 5: We thank the reviewer for pointing out the lack of information regarding the reliability and validity of the Smartphone Overdependence Scale. We have added this information, drawn from recent studies, and reported the metrics from our sample. In addition, we have added citations and minor discussions to improve our grounding in our construct grouping section.

Please refer to the Outcome section on pages 4-5.

The Smartphone Overdependence Scale has been validated in large, nationally representative adolescent samples. In the 2020 Korea Youth Risk Behavior Survey (n = 54,948), Cronbach’s α was 0.92 [47], and the original development study reported α = 0.84 [45], further supporting reliability across populations. Convergent validity has been demonstrated through consistent associations with stress, low sleep quality, unhappiness, sadness/despair, and loneliness—patterns theoretically aligned with overdependence [45]. The Cronbach’s α in the present study sample was 0.86.”

Please refer to points 1 and 2 in the Conceptual Feature Selection & Construct Grouping section on page 5.

“Intention to seek smartphone education can be understood as a proxy within the broader help- seeking framework for behavioral addictions. Treatment- or help-seeking studies highlight its association with smartphone struggles across countries [48–50] and in other smartphone- accessible issues such as internet gaming disorder, internet addiction, and social network sites [51,52].”

“Although evidence for their efficacy and effectiveness remains mixed [ 55 –58], adolescents who have received such education often demonstrate greater awareness and are more likely to engage in help-seeking behaviors. Those who have received such education tend to have greater awareness of the risks, perceive the programs as more helpful when at high risk, and even demonstrate reduced overdependence and improved self-control in some cases [53,59,60].”

Comment 6:

RESULTS

The results are clearly presented, with tables, figures, and the appendix effectively complementing the text.

Regarding gender, please consider the Instructions for Authors on the Journal’s website, which require:

  • description of whether sex and/or gender differences may be expected;
  • reporting how sex and/or gender were accounted for in the study design;
  • provision of disaggregated data by sex and/or gender, where appropriate;
  • discussion of respective If a sex/gender analysis was not conducted, a rationale should be provided in the Discussion.

Response 6: We thank the reviewer for the positive evaluation of the results. Also, we thank the reviewer for their attention to this journal policy, which we mistakenly overlooked. We have provided a short discussion on sex differences in the Limitations & Future Research section while clarifying that biological sex was collected.

Please refer to the last paragraph on page 20 in the Limitations & Future Research section.

“Although biological sex was accounted for as a demographic variable, sex-disaggregated analyses were not conducted in this study. Prior research in Korean adolescents has produced mixed findings regarding sex differences in smartphone overdependence, and no definitive pattern has been established. For example, Noh et al. [8] reported a higher risk among girls, but other studies did not emphasize or systematically examine sex differences [115]. Moreover, Chen et al. [116] found no significant sex differences in smartphone addiction prevalence among Chinese medical college students. Given this lack of consistent evidence and considering the study’s primary objective of developing an overall predictive model, subgroup analyses were not pursued. Future research should explore sex variations more directly and robustly.”

Comment 7:

DISCUSSION

The discussion appropriately addresses the findings in relation to previous studies and the study’s aims. Broader public health implications and limitations are noted, and future research directions are outlined. Suggested improvements:

Line 550–553:

Clarify the frameworks referenced in the statement: “It aligns closely with cognitive-behavioral frameworks by examining how adolescents evaluate and perceive their own ability and issues with smartphone use” (add specific citations).

Support with citations the claim that features related to perceived control, awareness of usage patterns, and beliefs about digital competencies correspond to mechanisms in models of behavioral addiction and cognitive distortions.

The references are generally adequate, but additional citations following the above suggestions would strengthen the manuscript.

Response 7: We thank the reviewer for their suggestions surrounding our discussion on how our results relate to the cognitive-behavioral framework. We have revised the discussion section regarding perceived digital competence and cognitive-behavioral theory with further discussion and references. We also appreciate the reviewer’s attention to detail regarding citations. We have added additional citations throughout our manuscript and feel that it has substantially strengthened our paper.

Please refer to the last paragraph on page 18 in the Discussion section.

“Cognitive distortions are systematic errors in thinking that bias perceptions of self, the world, or one’s control, and are central mechanisms targeted in cognitive-behavioral therapy [108]. The nonlinear and parabolic trends observed in this model’s PDPs suggest that adolescents’ cognitive self-perceptions are complex, potentially involving both denial and overconfidence, patterns well- documented in cognitive-behavioral theory [108 – 110] and often targeted in CBT-based interventions for addictive behaviors [111]. Notably, some indicators, such as overconfidence in digital literacy, were associated with increased risk, perhaps reflecting distorted cognitions or rationalizations, well-known patterns in CBT models that justify and maintain excessive use [ 84, 108], which merit further psychological investigation.”

Comment 8:

In conclusion, this is a timely study with a well-written manuscript, rigorous methodology, and valuable findings, especially given the novel analytical approach. While I recommend the article for consideration in the International Journal of Environmental Research and Public Health, major revisions are necessary.

I wish the authors success in revising and publishing their work.

Response 8:

We thank the reviewer for their time in reviewing and providing structured feedback. We believe that by incorporating these suggestions, our manuscript has been substantially strengthened.

Reviewer 2 Report

Comments and Suggestions for Authors

Congratulations team. I have a few suggestions for your consideration:

Line 26- Can you add a specific age range here?

Line 30- Add the full abbreviation of OCD and ADHD

Line 93-95- any reference would be great

Methods:

  • How was the sample size calculated?
  • How is the missing data handled?
  • Add specific exclusion and inclusion criteria.
  • Why was the age range of 10-18 years selected? Please justify the cutoff values.

  • Discussion: Can you make the policy more specific for metropolitan, small/medium cities, and rural areas, and how these results align with or differ from previous literature? Please mention it.

Author Response

Comment 1: Line 26- Can you add a specific age range here?
Response 1: We thank the reviewer for pointing out this missing detail and have included it. 
Please refer to the second paragraph in the Introduction section on page 2.
“The NISA report [7] estimates that 37% to 40% of Korean adolescents between the ages of 10-19 
meet criteria for smartphone overdependence, the highest among all age groups.”

Comment 2: Line 30- Add the full abbreviation of OCD and ADHD
Response 2: We thank the reviewer for pointing out this error. We have defined each acronym.

Please refer to the first paragraph in the Introduction section on page 2.
“These prevalence estimates are particularly noteworthy given consistent associations with 
depression, anxiety, obsessive–compulsive disorder (OCD), attention deficit hyperactivity disorder 
(ADHD), stress, loneliness, suicidal ideation, and substance use [16 –18].”

Comment 3: Line 93-95- any reference would be great.
Response 3: We thank the reviewer for their suggestion. We have added a reference and expanded on 
this section. 
Please refer to the third paragraph in the Introduction section on page 3. 
“Sapienza et al. [37] found that urban residents spend more total time on smartphones, often for 
productivity, whereas rural residents spend proportionally more time on entertainment. In 
Bangladesh and Korea, higher population density has been linked to elevated risk [38], while 
research in China found rural adolescents more vulnerable, citing loneliness and anxiety [ 39, 40 ].”

Comment 3: 
Methods:
• How was the sample size calculated?
• How is the missing data handled?
• Add specific exclusion and inclusion criteria.
• Why was the age range of 10-18 years selected? Please justify the cutoff values.
Response 3: 
We thank the reviewer for calling attention to the components missing in our methodology. We have 
revised our methods sections, clarifying that we included all available adolescent respondents aged 
10–18 years from our survey and justifying our age range. Next, we discuss the data processing steps 
taken to address missingness to achieve complete cases.
Please refer to the Data section on page 4.
“The NISA Smartphone Overdependence Survey is a nationally representative household survey 
of individuals aged 3–69 years in South Korea, based on probability sampling…In the current 
study, we include all 1,873 adolescent participants aged 10-18 from the full sample. The upper 
threshold of age 18 was selected since it marks the final year of high school in South Korea and 
avoids conflating adolescent experiences with those of emerging adults. College, typically age 19 
in Korea, has been shown to signal a profound shift in lifestyle, social environment, academic 
context, and resultant smartphone use patterns [42]. This age range is consistent with other Korean 
epidemiological and behavioral health studies demonstrating that smartphone use and its 
psychological consequences are most relevant within school-aged populations [16, 43, 44].”
Please refer to the Conceptual Feature Selection & Construct Grouping section on page 5.
“After one-hot encoding, we drop 14 features with missingness due to skip logic and one feature due to never being endorsed in the sample. In the case of skip logic missingness, other retained 
features were determined to provide conceptual overlap. For example, the feature “do you think 
the content you typically use on smartphone video streaming services is primarily entertainment?” 
would overlap with the retained binary indicator of “TV/Entertainment as the #1 mostly used 
video content". Other examples include Likert scale questions about their experience with 
smartphone preventive education. In these cases, a "0" fill approach would not be appropriate and 
would substantially restrict the sample (69% of adolescents never attended a preventive program). 
Instead, we retain the attendance feature. The final dataset resulted in a total of 92 survey items 
and 132 features after one-hot encoding.”

Comment 4: 
Discussion: Can you make the policy more specific for metropolitan, small/medium cities, and rural 
areas, and how these results align with or differ from previous literature? Please mention it.
Response 4: 
We thank the reviewer for their feedback on improving our policy discussion, as this is a critical 
component of our study. We have clarified our policy recommendations by first discussing the unique 
contexts of the place-stratifications, clarifying the policy recommendations for each, and discussing 
how they align or diverge from prior literature. 
Please refer to the Place-Based Policy Implication Profiles section on pages 19-20.
“We identify tangible implications that can guide place-based policy responses across regional 
contexts for the prevention and intervention of smartphone overdependence among adolescents. 
The NISA survey uses the classification of municipalities outlined in South Korea’s Local 
Autonomy Act to stratify urbanicity. The Act distinguishes municipalities by population size and 
administrative capacity, designating metropolitan cities (≥ 500,000 residents), medium/small cities 
(≥ 50,000 but < 500,000), and town/rural counties as distinct units of local governance [41]. This 
stratification is also aligned with high–middle–low accessibility clusters in South Korea’s social 
infrastructures, illustrating how access patterns shape opportunity structures [112]. Each distinct 
stratum experiences its own unique exposures and consequences, requiring tailored preventive 
strategies. This approach is consistent with prior literature on spatial inequalities. For instance, 
Parker et al. [113] showed that urban, suburban, and rural communities differ in lived experiences 
in ways that extend beyond a simple binary, underscoring the need for nuanced stratification. By 
linking smartphone overdependence to broader discussions about urbanicity differences, our 
place-based profiles extend existing policy discussions toward more context-sensitive responses 
that can be readily utilized by the South Korean government.
… 
Rural districts are sparsely populated, with limited broadband infrastructure and fewer specialized 
health services [114]. At-risk adolescents in towns and rural areas predominantly use smartphones 
for expressive and leisure-related activities such as video streaming and engagement in online social interactions in the present study. These patterns are consistent with Sapienza et al. [37], who 
noted that rural users disproportionately use smartphones for entertainment, likely due to limited 
offline recreation opportunities. Similar rural-specific vulnerabilities have been documented in 
Bangladesh, where urbanicity predicted higher smartphone addiction in children, and in China, 
where rural adolescents reported greater dependence due to isolation and anxiety [ 39 ,40]. South 
Korean data further suggest that overdependence in rural adolescents is associated with anxiety 
and psychosocial difficulties [18]. These findings underscore that rural smartphone use often 
reflects structural constraints rather than mere individual preference. Accordingly, rural policies 
must go beyond limiting screen time to address structural constraints, such as expanding offline
youth facilities, strengthening community-driven recreation, and establishing safe online 
interaction guidelines. As Song suggested, telehealth services should also be prioritized to 
compensate for the lack of in-person resources, alongside school-linked screening and 
parent/teacher training to bridge prevention gaps [114]. Finally, because stress coping emerged as 
a prominent predictor of smartphone overdependence, interventions should actively incorporate 
adolescents’ input on their daily smartphone use to inform the development of meaningful and 
accessible offline alternatives."

Round 2

Reviewer 1 Report

Comments and Suggestions for Authors

I am pleased to see the improvements in this manuscript. It presents interesting data that are now better contextualized within the scope of the study. Therefore, I recommend that the Editor(s) consider this paper for publication in the prestigious journal IJERPH.